# Alignment Based Matching Networks for One-Shot Classification and Open-Set Recognition

## Abstract

Deep learning for object classification relies heavily on convolutional models. While effective, CNNs are rarely interpretable after the fact. An attention mechanism can be used to highlight the area of the image that the model focuses on thus offering a narrow view into the mechanism of classification. We expand on this idea by forcing the method to explicitly align images to be classified to reference images representing the classes. The mechanism of alignment is learned and therefore does not require that the reference objects are anything like those being classified. Beyond explanation, our exemplar based cross-alignment method enables classification with only a single example per category (one-shot). Our model cuts the 5-way, 1-shot error rate in Omniglot from 2.1% to 1.4% and in MiniImageNet from 53.5% to 46.5% while simultaneously providing point-wise alignment information providing some understanding on what the network is capturing. This method of alignment also enables the recognition of an unsupported class (open-set) in the one-shot setting while maintaining an F1-score of above 0.5 for Omniglot even with 19 other distracting classes while baselines completely fail to separate the open-set class in the one-shot setting.

## 1 Introduction

Convolutional neural networks (CNNs) in various realizations have come to dominate image, speech, and even text processing. Performance gains afforded by such architectures, with learned, multi-stage image features, emerge from large amounts of supervised training data. In addition to being data hungry, the networks are also challenging to interpret after the fact, potentially yielding confident yet unwarranted predictions for tailored examples Yosinski et al. (2015b); Szegedy et al. (2013). Our goal in contrast is to learn from very few examples, and specifically tailor the models to emphasize interpretability.

Deep, complex architectures, despite large numbers of parameters involved, are not incompatible with one-shot or open-set learning. Instead of predicting categories directly, one can set up the learning problem to recognize sameness between pairs of images as in Siamese networks Koch et al. (2015) or between a set of support images as in Matching Networks Vinyals et al. (2016). In this case, a relatively large dataset of pairs is required to help the network learn features that summarize images in a manner that is tailored for assessing category differences. Such a network can succeed in predicting sameness for entirely new categories or comparing new images to a set of given reference sets. We depart from this view by including a learned alignment step in the comparison. Each new image is aligned to a reference image and it is the resulting learned alignment score that guides the selection of which reference image it is most related to.

Our approach of learning to align images to reference objects gives a degree of freedom to select reference objects that may differ substantially from the test images (by shape, style or even type). By construction, the resulting alignment is also interpretable about the relation, and can be examined after the fact. The explicit alignment also allows a channel of feedback for learning beyond the label classification. We learn multi-scale image features, possibly different representations for reference and target images, and use them to learn point-to-point matches. These point matches which, are known for a self-alignment (aligning an image to itself), can be used to learn with a much higher

bandwidth signal than the traditional label-only signal. The method is specifically geared towards one-shot learning where a new set of reference objects are given, one example per category, and the new images are to be classified accordingly without additional labels. As our method computes an alignment between images, we call our model Alignment Based Matching Networks (ABM Nets).

We demonstrate that the matching images through ABM Nets can yield state-of-the-art accuracies for the one-shot learning task on Omniglot and MiniImageNet datasets. Further, we show that ABM networks outperforms these other state-of-the-art models on the open-set recognition task while in the one-shot setting by achieving a high accuracy of matching the non-open-set classes while maintaining a high F1 score for identifying samples in the open-set. Additionally, we show how ABM Nets provide for free, some understanding into why our network selects a label for a target image by querying individual point match likelihoods for pairs of points selected between the target and support images. Point matchings allow us to quickly determine the strength of the semantic information learned by the network. We contrast the strength of the semantic information learned for handwritten digit recognition to the more superficial information learned for real world images.

Our main contributions can be summarized as follows:

- We cast the one-shot learning task as the outcome of an alignment task and extend the Matching Network model for learning these explicit alignments
- We introduce self-alignment based regularization as a mechanism of providing a high-bandwidth signal to an otherwise low-bandwidth task
- We demonstrate state-of-the-art performance on the one-shot learning task for Omniglot and MiniImageNet tasks
- We show how the same ABM network architecture can be used to perform open-set recognition in the one-shot setting, outperforming existing alternatives

The next section describes related work. We will then introduce the ABM model and then evaluate the model on the one-shot learning and one-shot open-set recognition tasks.

## 2 RELATED WORK

Matching points from one image to another has been traditionally performed with robust descriptors such as SIFT Lowe (1999), HOG and their variants. A neural network can even be trained to learn the descriptor and orientation from dataYi et al. (2016) or to detect keypoints for patch-matching Altwaijry et al. (2016). These methods enable us to identify distinctive objects under many different conditions such as affine transformations, orientation changes, different illuminations, etc. These descriptors can however, be sensitive to different realizations of an object type making them less suitable for object classification than CNNs.

CNNs have dominated the visual classification tasks over the past few years and have grown increasingly complex and deep. Understanding why these networks produce specific results has not been an easy task. Most venues of explanations are provided either through visualizing the activation of various layers of a network or by generating images from the network Zeiler & Fergus (2014); Mahendran & Vedaldi (2015); Yosinski et al. (2015a). These give us insight into properties of the network such as the areas of visual attention of the network or types of properties captured by those nodes. However, the correspondence between parts of images and finer grained information is lost as we go to the higher layers of the network.

Instead of finding correspondences between parts of images, image pairs can be directly embedded into a space where their similarity can be queried. Such models are used in one-shot learning settings to match pairs of images from previously unseen tasks. Networks like the Siamese neural networks Koch et al. (2015) encode images into a feature representation given by the top layer of a CNN and compute similarity in the CNN space. Matching networks Vinyals et al. (2016) take this a step further by comparing sets of labeled support images to unlabeled target images in the top level feature encoding space to find the closest match among the support images. They also design a training procedure to specifically train networks for one-shot learning. This has been extended by Prototypical Networks Snell et al. (2017) to use a prototype per class, corresponding to the mean embedded vector per class, rather than the mean distances to the embedded vectors to improve the performance of these networks in the few-shot learning setting.

Finding the right training procedure specifically for one-shot learning has been abstracted out in meta-learning schemes which learn how to perform gradient descent to best optimize the models and how to best initialize them to good starting points. Meta-learning LSTMsRavi & Larochelle (2017) use LSTMs to predict parameter updates while Temporal Convolution based Meta Learners Mishra et al. (2017) use deep recurrent networks based on dilated convolutions for few-shot learning tasks. The meta-learning framework is quite powerful as it can be used as a means of rapid adaptation of deep nets in a model-agnostic way.

Attentive Recurrent ComparatorsShyam & Dukkipati (2017) frame the one-shot learning task using an LSTM that takes multiple "glimpses" of a pair of images alternatively and repeatedly updating the hidden state of an RNN controller. The final hidden state of the controller is used to determine the similarity between pairs of images.

The one-shot learning task can be made even more challenging when the test image does not correspond to any of the reference classes. The test image in this case belong to the open-set and the task of identifying such images is the open-set recognition task. Methods to solve the general open-set recognition task (in the non-one-shot learning setting) involve two primary components - inducing a distance metric over objects such that objects belonging to the same class have low distances and then establishing a Compact Abated Probability (CAP) model Scheirer et al. (2014) over these distances such that when the probability falls below a threshold for all known classes, then the object is determined to be in the open set. Weibull-calibrated SVMs (W-SVM) Scheirer et al. (2014) use support vector machines to learn a kernel function with which distances are computed and extreme value theory to establish the CAP model based on the distribution of distances obtained from the known classes. Sparse representation-based open-set recognition (SROSR) Zhang & Patel (2017) extend the sparse representation-based classification algorithm Wright et al. (2009) learns a sparse representation based on reconstruction errors and build a CAP model on the reconstruction error distribution to establish the open-set threshold. Bendale & Boult (2016) bring CAP models to deep learning models using a generalization of the softmax operator, the open-max, to learn the threshold for the open set.

## 3 METHOD

We present Alignment Based Matching Networks (ABM Nets) - a two tier matching model to match images via alignment. The first tier parameterizes the probability that a pixel in the test image matches a pixel in a reference image by forcing an alignment between the pair of images and computing an alignment score. The second tier aggregates the pixel alignment likelihoods between the test image and a set of reference images and uses the aggregate score to find the likelihood that the test image matches each of the reference images.

### 3.1 POINT MATCHING VIA ALIGNMENT

Formally, classifying a target image $I_t$ given a set of support images $S = \{(I_k, y_k) : i \in \{1 \dots k\}\}$, where $y_k$ is the label for image $I_k$, corresponds to predicting the label $y_t$ which maximizes $P(y_t|I_t, S)$. To align images, we first look pairwise matchings $P(y_t = y_k|I_t, I_k)$. Traditionally, this function is approximated using a neural net with parameters $\theta$ as

$$P_\theta(y_t = y_k|I_t, I_k) \propto exp(D(f_\theta(I_t), g_\theta(I_k)))$$

Where $f = g$ are the top layers of a CNNs that embed the images into a common space where a distance function can be applied (such as cosine similarity).

While this provides us a straightforward mechanism to match images, it treats the CNN as a black box and does not yield any insight into correspondences between images. Two examples from the same class can not be aligned in a straightforward way as the finer resolution information is lost as the network architecture gets deeper.

To overcome this problem, we propose classifying images through an alignment rather than a full embedding of the image. We assume that there is an alignment $M(I_t, S) \in \mathcal{M}$ which matches points from $I_t$ to $S$, but is not directly observable to us, where $\mathcal{M}$ is the set of all possible image alignments of the test image to the reference image set. We express the alignment $M(I_t, S)$ as a binary tensor $M_{i,j,k}(I_t, S)$ which is 1 when pixel $i$ in $I_t$ matches to pixel $j$ in $I_k \in S$. For brevity,

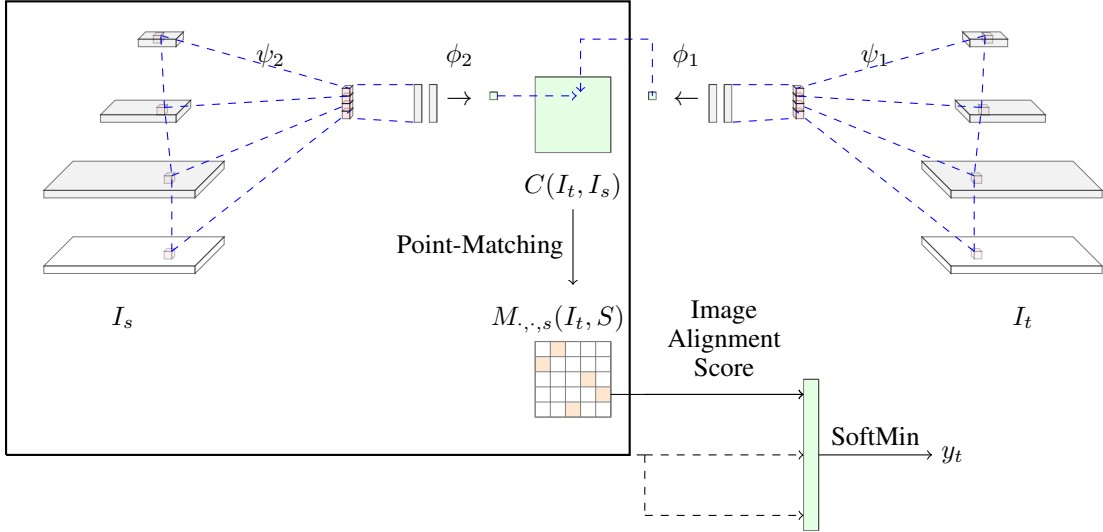

Figure 1: Tier 1 of the matching system uses two CNN encoders used to generate the point-wise matching costs $C(I_t, I_s)$. The second tier computes the net alignment costs to select the matching label.

we write $M(I_t, S) = M|I_t, S$. We can now express the probability assigning the label $y_k$ associated with image $I_k \in S$ to the rest image as

$$P(y_t = y_k | I_t, S) = \sum_{M \in \mathcal{M}} P(y_t = y_k | M, I_t, S) \cdot P(M | I_t, S)$$

This formulation captures how closely two images are related to each other through the individual point-wise mapping. However, marginalizing over all possible matchings is intractable.

To make the objective tractable, we make two main approximations. First, we approximate each point mapping as an independent indicator of the alignment, we get $P(M|I_t, S) = \prod_i P_\theta(M_{i,\cdot,\cdot} = 1 | I_t, S)$. Second, we assume that the correct alignment $M^*$ is much more likely than incorrect ones giving

$$P(y_t = y_k | I_t, S) \approx \prod_i \max_j P(M_{i,j,k} = 1 | I_t, S) \tag{1}$$

As the label $\hat{y}_t = \max_k P(y_t = y_k | I_t, S)$, the ABM model becomes a two-tier matching scheme - first a matching to align pixels between the images and second to match the label to the best aligning image. Figure 1 visually illustrates the two tier matching.

By finding the matching across all support images at once, the individual pixel mapping learns to better contrast the matchings across various reference images. Expressing the cost of setting $M_{i,j,k} = 1$ as $C_{i,j}(I_t, I_k)$, we express the probability of a matching $M$ as an independent point matching across the pixels in the test image as

$$P(M_{i,j,k} | I_t, S) \propto exp(-C_{i,j}(I_t, I_k)) \tag{2}$$

We compute the alignment cost as distances in an embedded space $C_{i,j}(I_t, I_s) = D(\phi(I_t, i), \phi(I_s, j))$. As just the plain RGB values do not convey any semantic notions such as object type or shape, we use a hyper-column descriptor Hariharan et al. (2015) using a CNN with parameters $\theta$ to learn the pixel embedding $\phi = \psi_\theta(I., i)$. The initial few dimensions of this feature space are designed to provide high resolution information of the pixels which is needed for fine-grained alignment while the remaining dimensions are designed provide increasingly more contextual information useful for distinguishing between pixels that would other wise be identical without context. In our experiments, we use cosine distances as the distance measure to be consistent with Vinyals et al. (2016).

Note that we can compare dissimilar test and reference images (such as gray-scale reference images and RGB test images of different sizes) by using separate encoders $\phi_1, \phi_2$. Also, note that the greedy matching can result in multiple points in the test image matching to the same point in the reference image. If the matching has to be unique, one can use the Hungarian algorithm Kuhn (1955) (or its GPU capable counterpart, the Auction algorithm Vasconcelos & Rosenhahn (2009)).

Finally, we note that in order to perform the alignments, we do require additional computation. The comparison phase of Matching Networks Vinyals et al. (2016) when encoding to a $d_L$ dimensional feature space takes $O(d_L n_s)$ time for comparing against $n_s$ reference images. On the other hand, a full pixel-to-pixel alignment on images of size $m \times n$ with ABM Networks takes time $O(mnn_s(\sum_l d_l))$. For both models, encoding the image using a CNN takes $O(mnn_s d_1)$ time for computing the first hidden layer. Hence, even with the pixel-wise matching, the asymptotic runtime complexity remains the same. In practice, the constant factor multiplier can make aligning all points relatively slow. In our experiments, we found that matching $10\%$ of the test image to $20\%$ of the reference image allows us to maintain high performance without impacting the total runtime significantly.

## 4 OPEN SET RECOGNITION

To extend the alignment based matching framework for one-shot open-set recognition, we first need to define the distance between two images. Combining Equations 1 and 2, we have

$$P(y_t = y_k | I_t, S) \propto exp\left(-\sum_i \min_j C_{i,j}(I_t, I_k)\right)$$

We can leverage this to define the distance between two images $\zeta(I_t, I_s)$ as the sum of the point-to-point alignment costs $\zeta(I_t, I_s) = \sum_{i \in I_t} \min_{j \in I_s} C_{i,j}(I_t, I_s)$

By using the OpenMax Bendale & Boult (2016) classification strategy, we can learn the distance threshold $\tau$ that determines if the test image is matched to a reference set image or to the open set as

$$P(y_t = k | I_t, S) = \begin{cases} \frac{exp(-\zeta(I_t, I_k))}{exp(-\tau) + \sum_s exp(-\zeta(I_t, I_s))} & y = k \\ \frac{exp(-\tau)}{exp(-\tau)\sum_s exp(-\zeta(I_t, I_s))} & y \in \text{open set} \end{cases}$$

When $\zeta(I_t, I_k) > \tau \forall k$, we produce the label $y = 0$ corresponding to the open set. Otherwise, the label is selected as $y_k = \arg\min_k \zeta(I_t, I_k)$.

### 4.1 SELF-REGULARIZATION AND TRAINING STRATEGY

During training, we additionally compute self-regularization by aligning the test image to itself. The self-regularization loss $\ell_{\text{self}}(I_t, \theta)$ is computed by computing the categorical loss of $P(M_{i,j,k} | I_t, \{I_t\})$ over the identity mapping.

We train the model directly for one-shot open-set recognition via simulations. A task $\mathbb{T}$ is defined as a (uniform) distribution over possible label sets $L$. A trial is formed by first sampling $L$ from $\mathbb{T}$. Next, the open set label $l_o$ is sampled from $L$. We then sample a reference set $S$ from $L \setminus \{l_o\}$ and test images $T$ from $L$. The network is then trained to minimize the error predicting the labels of the test images $T$ conditioned on the reference set $S$.

The overall training objective is given by

$$\theta = \arg\max_\theta E_{L \sim \mathbb{T}}\left[E_{l_o \sim L}\left[E_{S \sim L \setminus \{l_o\}, T \sim L}\left[\sum_{(I,y) \in T} \log P_\theta(y | I, S) + \ell_{\text{self}}(I, \theta)\right]\right]\right]$$

For the standard one-shot learning task, $l_0 = \phi$, and the open-set sampling phase is omitted and we obtain the one-shot learning training strategy used in Vinyals et al. (2016).

| Model | MNIST 5-way Acc | | CIFAR10 5-way Acc | |
|---|---|---|---|---|
| | 1-shot | 5-shot | 1-shot | 5-shot |
| Matching Networks (no FCE) | 69.3% | 83.9% | 46.7% | 54.2% |
| ABM Networks (Ours) | 72.7% | 90.3% | 45.4% | 53.9% |
| ABM + Self Reg | 79.6% | 93.3% | 49.4% | 58.0% |

Table 1: One-shot learning on MNIST and CIFAR10 datasets compared to matching networks. Identical CNN encoders are used for both models.

## 5 EXPERIMENTS

We first test ABM Nets on MIST and CIFAR10 datasets to demonstrate how these networks generalize better with limited training classes. We then train the model on Omniglot and MiniImageNet datasets to compare against current state-of-the-art methods for one-shot learning. We finally test our model on the one-shot open-set recognition task. Our main baseline is Matching Networks without Fully Contextual Embeddings (FCE) as we use the same encoder architecture, same number of parameters and same learning algorithm for the most fair comparison.

As our focus is on the model rather than the training scheme, we will not use meta-learners to optimize or initialize the parameters of the model. Instead, we use stochastic optimization via AdamKingma & Ba (2015) to train our model and Xavier uniform initialization.

### 5.1 ONE-SHOT LEARNING ON SMALL DATASETS

The MNIST and CIFAR10 have only 10 classes each in the dataset. We use 5 classes for training and the other 5 for validation and testing. As only 5 classes are available during training, it is easy for networks that embed the entire image without alignment to overfit on the training classes.

The datasets are augmented using random rotations by multiples of 90 degrees. Images are rescaled to $28x28$. For our model we use a 4 layered CNN with $32, 64, 64, 64$ filters - identical to that used in Matching Networks for their Omniglot experiments, ensuring that both models have the same number of parameters. All layers have ReLU nonlinearities and are trained with batch normalization. 10% of the test image pixels are uniformly sampled and matched to a uniform sample of 20% of the reference image pixels. The point matching cost is computed using negative cosine similarity between the stacked feature vectors. The alignment between the reference and test image is computed as the average independent (greedy) pixel-wise minimum cost matching. For the baseline, we use cosine similarities between the embedded space and we do not use Fully Conditional Embeddings (FCE).

Training is done over 1000 batched training trials each epoch over 200 epochs with batch running 32 one-shot trials . Validation and testing use 500 and 1000 batched trials correspondingly. Optimization is done using Adam with a weight decay of $10^{-4}$ and learning rate decay of $10^{-6}$.

ABM Networks generalize much better than Matching Networks on MNIST and CIFAR10 datasets as shown in Table 1. On the 5-way, 1-shot learning task, ABM Networks achieve an accuracy of 72.7% on MNIST when compared to the 69.3% of matching networks. With self-regularization, this accuracy jumps to 79.6% demonstrating how self-regularization helps prevent overfitting. On the CIFAR10 dataset, ABM Nets with self-regularization yields an accuracy of 49.4% over 46.7% of Matching Nets and without self-regularization, ABM Nets perform slightly worse than Matching Nets showing that self-regularization is a powerful tool for real images as well as simple digit ones.

In addition to one-shot classification, ABM Nets allow us to align images for free as a by-product of the classification task. Figure 2 shows the alignment probabilities $P(M|I_t, I_s)$ for pairs of test and reference images for randomly sampled pixels in the test image. We can see that without self-regularization, the uncertainty of alignment is much higher showing that the hyper-column filters are much better at producing identifiable pixels. We can see that the alignment provides us a way to gain insight into what the model is learning.

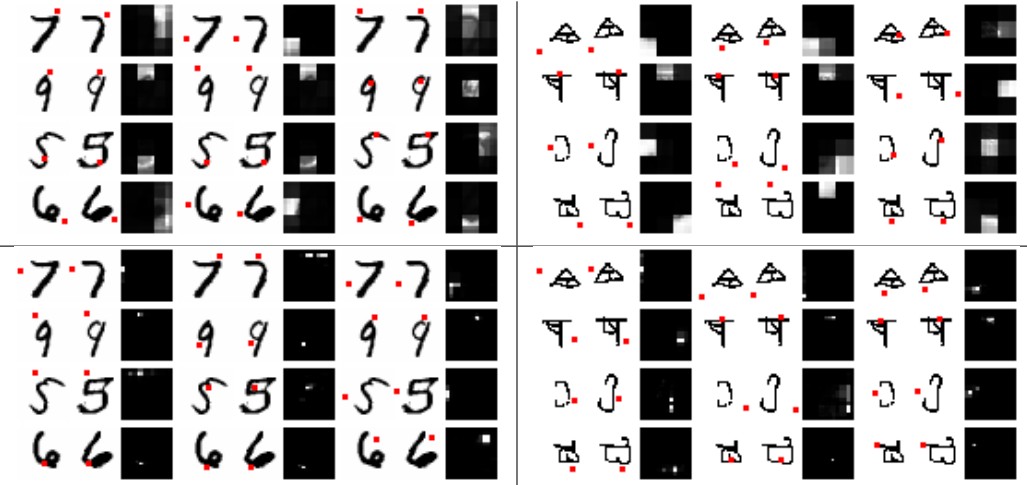

Figure 2: Three points sampled uniformly from the test image (columns 1, 4,7) are mapped to the reference image (columns 2,5,8) using the MNIST dataset (left) and Omniglot dataset (right). The red point in the test image is the point selected for matching. The red point in the reference image shows the corresponding minimum cost matching. The matching probability (columns 3,6,9) is obtained by matching the selected point to all points in the reference image. Each row show a different test-reference image pair. Results are shown for ABM Nets without (top) and with (bottom) self-regularization

## 5.2 ONE-SHOT LEARNING WITH OMNIGLOT

We test ABM Nets on the Omniglot dataset, a dataset more geared towards the one-shot learning task using the same model structure as the MNIST task. Of the 1623 classes in this dataset, 1200 are used for training, 300 for testing and the remaining for validation. Images are resized to $28 \times 28$. Training and testing procedures are identical to the previous task. We use a batch size of 32 for the 5-shot learning task, but use a batch size of 4 for the 20-way task due to GPU memory constraints. This CNN is identical to Vinyals et al. (2016).

Table 2 shows the results for 5-way and 20-way 1-shot and 5-shot learning tasks. We see that ABM Nets achieve an accuracy of 98.6% up from 97.9% of matching networks. For the 20-way, 1-shot learning task, our model achieves an accuracy of 96.5% compared to 93.5% of Matching Networks. Naive and Convolutional Attentive Recurrent ComparatorsShyam & Dukkipati (2017) both achieve lower accuracies while Full-Context Convolutional ARC beat ABM Nets with an accuracy of 97.5% but while using a much more complicated model with more parameters. It must also be noted that experimental conditions in ABM Nets and Matching Networks vary slightly from ARCs (such as the use $32 \times 32$ images and data augmentation with affine transforms). We use an identical setting to Matching Networks Vinyals et al. (2016) ($28 \times 28$ images, 90 degree multiple rotations and same number of parameters) so as to draw a fair comparison with the baseline model.

The point matchings shown in Figure 2 shows the reasonable matchings produced for random points from both the foreground and background between the test and reference image. It must be noted that no spatial constraints are used for matching. We can see that similar to the matching distribution in MNIST, self-regularization provides sharper matchings indicating that the hyper-column filters are more identifiable.

## 5.3 ONE-SHOT MINIIMAGENET

The MiniImageNet dataset is a subset of the ImageNet Deng et al. (2009) dataset consisting of 100 classes with 600 examples each. The classes are divided into 64 training, 16 validation and 20 testing classes using the same class splits used in Meta-learning LSTMsRavi & Larochelle (2017). Images are of size $84 \times 84$. While a more complicated model can be chosen as an encoder (such as Inception Net or even VGG variants with more filters), we keep the same structure as presented for

| Model | 5-way Acc | | 20-way Acc |
|---|---|---|---|
| | 1-shot | 5-shot | 1-shot |
| Matching Nets Vinyals et al. (2016) | 97.9% | 98.7% | 93.5% |
| Prototypical Nets Snell et al. (2017) | 98.8% | 99.7% | 96.0% |
| MAML Finn et al. (2017) | 98.7% | 99.9% | 95.8% |
| Meta Networks Munkhdalai & Yu (2017) | 98.9% | - | 97.0% |
| Naive ConvARC Shyam & Dukkipati (2017) | - | - | 96.1% |
| Full context ConvARC Shyam & Dukkipati (2017) | - | - | 97.5% |
| TCML Mishra et al. (2017) | 99.0% | 99.8% | 97.6% |
| ABM Nets (Ours) | 98.5% | 99.6% | 95.2% |
| ABM Nets + Self Reg | 98.6% | 99.8% | 96.5% |

Table 2: One-shot learning on Omniglot dataset

| Model | 5-way Acc | |
|---|---|---|
| | 1-shot | 5-shot |
| Matching Networks (no FCE) Vinyals et al. (2016) | 42.4% | 58.0% |
| Matching Networks (FCE) Vinyals et al. (2016) | 46.6% | 60.0% |
| Prototypical Networks Snell et al. (2017) | $49.42 \pm 0.78\%$ | $68.20 \pm 0.66\%$ |
| Meta Networks Munkhdalai & Yu (2017) | $49.21 \pm 0.96\%$ | - |
| MAML Finn et al. (2017) | $48.70 \pm 1.84\%$ | $63.15 \pm 0.91\%$ |
| TCML Mishra et al. (2017) | $55.71 \pm 0.99\%$ | $68.88 \pm 0.92\%$ |
| ABM Networks (Ours) | $48.92 \pm 0.69\%$ | $57.98 \pm 0.77\%$ |
| ABM + Self Reg | $52.54 \pm 0.70\%$ | $63.05 \pm 0.72\%$ |
| ABM + Self Reg (layers 4-6) | $53.47 \pm 0.67\%$ | $61.82 \pm 0.73\%$ |

Table 3: One-shot learning on MiniImageNet dataset compared to matching networks.

Matching NetworksVinyals et al. (2016) so as to have a fair grounds for comparison. The encoder is a 6 layered network with $32, 64, 64, 64, 64, 64$ sized filters in the corresponding layers. We sample 10% of the template image pixels and match them to 20% of the reference image pixels just as in the other experiments. We use a batch size of 32 and the same optimizer settings for Adam as the other experiments. Batch normalization and ReLU activations are used.

ABMs outperform the Matching Network baseline (Table 3) improving the accuracy from 42.4% to 51.3% for 5-way, 1-shot learning task and from 58.0% to 63.0% for the 5-way, 5-shot learning task. with self-regularization. Self-regularization out-performs one-shot learning with Matching Networks with Fully Contextual Embeddings while using a simpler model with fewer parameters. As there is much more variability in real world images, we also consider computing the alignment score using only the top half of the layers which improves the one-shot accuracy to $53.5\%$.

While accuracies improve, Figure 3 shows a stark contrast in the point mapping quality between the handwritten digit datasets and MiniImageNet. Texture and some simpler such features play a clear role in the matching, however, an overall semantic understanding of the scene seems to be lacking from the features extracted by this network. This suggests that while more elaborate meta-learning training schemes may squeeze a bit more accuracy out of such simple models stacked models, there is a more fundamental gap between the models and their ability to extract the semantic information needed for 1-shot learning as these models quickly adapt to simple texture information.

## 5.4 ONE-SHOT OPEN-SET RECOGNITION ON MNIST

We use the MNIST dataset to perform an N-way, one shot learning task with N-1 of the reference classes having one reference image each and the last class with no support (the open set). In other words, $|L| = N$ and $|S| = N - 1$. Classes 0-4 are used for training and 5-9 for testing and validation.

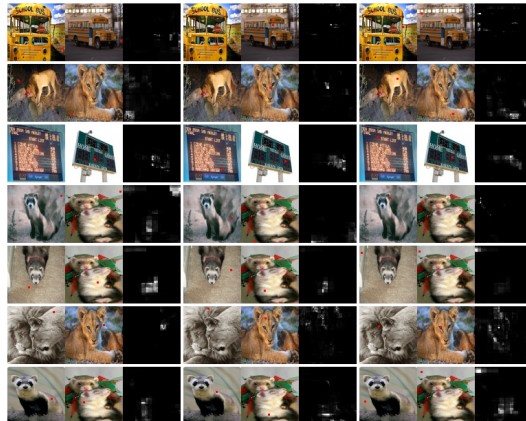

Figure 3: MiniImageNet point matching alignments. Best viewed with magnification.

| Model | $N = 2$ | | $N = 3$ | | $N = 4$ | | $N = 5$ | |
|---|---|---|---|---|---|---|---|---|
| | Acc | F1 | Acc | F1 | Acc | F1 | Acc | F1 |
| Matching Nets | 70.4% | 0.615 | 63.8% | 0.316 | 60.1% | 0.172 | 58.8% | 0.101 |
| ABM Nets | 72.2% | 0.650 | 67.6% | 0.443 | 63.2% | 0.160 | 63.1% | 0.197 |
| ABM Nets + Reg | 71.6% | 0.643 | 66.9% | 0.404 | 64.7% | 0.177 | 65.5% | 0.164 |

Table 4: Open set recognition in a one-shot learning setup on MNIST dataset. Of the $N$ classes, $N - 1$ have reference with the last one being the open set. Matching via alignment yields better generalization accuracies and F1 scores

We train the model using the one-shot open-set simulation strategy. Hence, during testing, all the 5 classes reference images are sampled from are previously unseen classes during training, and one of these new classes is selected at random to have belong to the open set (is held out from the one-shot episode).

As a baseline, we use a Matching Network with a learned threshold similar to ABM Nets by modifying the softmax layer to be an openmax layer. We compute to metrics to evaluate the performance of the networks: classification accuracy and F1 score of recognizing the open set. The accuracy is the percent of labels correctly predicted, with label 0 being the correct label when the test image is in the open set. The corresponding F1 score is calculated using the precision and recall of the models for the binary classification problem of determining if the test image is in the open set or not. We use a batch size of 32. We use cosine distances for both ABM Nets and Matching Nets. The best performing model that is used for testing is selected using cross-validation as is done for the one-shot learning task.

Table 4 shows the results for the open set recognition task with one sample per class for $N = 2, 3, 4, 5$. $N = 2$ is a binary classification task which determines whether the test image matches the reference image. ABM Nets generalize much better to the test set yielding higher F1 scores and accuracies. For the 5-way classification task with 4 reference classes and one open-set, ABM Nets yields an accuracy of $65.5\%$ over the $58.8\%$ for the baseline with an F1 score of $0.164$ over the $0.101$ for Matching Nets.

## 5.5 One-shot open-set recognition with Omniglot

The variability across languages and how different people draw characters provides a better testing platform for evaluating the one-shot open-set recognition task. As we have more classes, we can test the effect of detecting the open set class with $N = 5, 10, 15$ and $20$. The same train-validation-test class splits used in the one-shot learning task are used for the open-set recognition task as well.

| Model | $N = 5$ | | $N = 10$ | | $N = 15$ | | $N = 20$ | |
|---|---|---|---|---|---|---|---|---|
| | Acc | F1 | Acc | F1 | Acc | F1 | Acc | F1 |
| Matching Nets | 86.2% | 0.599 | 86.2% | 0.107 | 88.4% | 0.003 | 88.4% | 0.000 |
| ABM Nets | 83.0% | 0.406 | 87.0% | 0.075 | 88.8% | 0.019 | 89.0% | 0.017 |
| ABM Nets + Reg | 90.9% | 0.759 | 93.1% | 0.678 | 93.3% | 0.587 | 93.3% | 0.504 |

Table 5: Open set recognition in a one-shot learning setup on Omniglot dataset. Of the $N$ classes, $N - 1$ have reference with the last one being the open set. Matching via alignment yields better generalization accuracies and F1 scores

As a baseline, we use Matching Networks with the learned distance threshold below which the test image needs to match a reference image to not be considered as part of the open set. We then test the one-shot open-set recognition problem on ABM networks both with and without self regularization. Note that we do not compare our model against standard open-set recognition models using extreme value theory Scheirer et al. (2014); Zhang & Patel (2017); Bendale & Boult (2016) as these models rely on distributions of distances to be available which is not available to us in the one-shot version of this task. The network architecture and training details are identical to Section 5.2.

Table 5 shows the results for the one-shot open-set recognition task on the Omniglot dataset. We can immediately observe that the self-regularization plays a large role improving the accuracy of the model in this low-data setting. Without self-regularization, Matching Networks performs approximately as well as ABM networks and even outperforms ABM networks for the $N = 5$ with an accuracy of $86.2\%$ and an F1 score of $0.599$ versus the $83.0\%$ of unregularized ABM nets which generates an F1 score of $0.406$ on the same task. However, as $N$ increases we see that Matching networks is unable to distinguish images in the open set with the F1 score sharply falling to $0.000$ for $N = 20$. ABM networks on the other hand perform better with an F1 score of $0.017$ for $N = 20$ while also achieving a slightly better accuracy of $89.0\%$ vs $88.4\%$ of Matching Networks.

With self-regularization, we see a significant improvement in classification accuracy, but more importantly, we see a dramatic improvement in the F1 score. The F1 score for ABM networks with regularization for $N = 5$ is at $0.759$ over the $0.599$ achieved by Matching Networks. As $N$ increases to 20, we see that the F1 score falls very slowly in comparison to Matching networks resulting in an F1 score of $0.504$ when compared to $0.017$ without regularization or $0.000$ of Matching networks. This ability to distinguish members of the open-set is not achieved by sacrificing the one-shot learning accuracy. ABM networks with regularization have an accuracy of $90.9\%$ for $N = 5$ and $93.3\%$ for $N = 20$ compared to the $86.2\%$ and $88.4\%$ accuracies achieved by Matching Networks on the same tasks.

## 6 CONCLUSION

We have demonstrated a two-tiered matching system that is first able to align images by match contextual points from one image to another and utilize the alignment to determine the object class. Further, we have demonstrated that this model is able to operate in the low-data setting of one-shot on handwritten datasets and real word images. In addition to providing state-of-the-art accuracies in the matching tasks, we also show how individual point alignment can be extracted naturally from our model which takes a step in the direction of understanding why the model picks various matchings.

The image classification task is inherently provides low feedback for training with a high input dimensionality. The alignment phase opens the model up a much higher bandwidth feedback to be used for training via self-regularization by learning to align images to themselves. This feedback channel is otherwise not available to black-box image encoders.

Finally, though performing alignments is more computationally expensive than a black-box encoding architecture of the same size, we show that state of the art performance is achievable without impacting the runtime. This is achieved through aligning a random subset of the pixels. Even a small subset, such as 10% of pixels, as used in our experiments, is sufficient to train the ABM model effectively.

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
