# OpenReview forum: "Alignment Based Mathching Networks for One-Shot Classification and Open-Set Recognition"
_ICLR.cc/2019/Conference_

### Official Review · AnonReviewer2 · 2018-10-31
**Review for ABM-Nets**

**Rating:** 4
**Confidence:** 4

**Review:**

In this work, the authors tackle the problem of few-shot learning and open-set classification using a new type of NNs which they call alignment-based matching networks or ABM-Nets for short. They main idea is to benefit from binary maps between the query image and the support set (for the case of few-shot learning for the sake of discussion here) to guide the similarity measure.

I have quite a few concerns;

- After reading the paper two times, I still couldn't find a clear explanation as how the binary map C is constructed. The paper says the cost of M,i,j,k = 1 is C. So what exactly happens given I_t and I_k. My understanding is that a vector representation of each image is obtained and then from those representations the matrix C is constructed (maybe an outer product or something). This does not come out clearly.

- Nevertheless, I suspect if such a construction (based on my understanding) is the right approach. Firstly, I guess the algorithm should somehow encourage to match more points between the images. Right now the loss  does not have such a term so hypothetically you can match two images because they just share a red pixel which is obviously not right.

- Aside from the above (so basically regularizing norm(C) somehow), one wonders why matching a point to several others (as done according to matrix C) is the right choice.

- Related to the issues mentioned before, I may also complain that matching far away points might not be ideal. Currently I do not see how this can be avoided nor a solid statement as why this should not be a problem.


- Another comment is how the alignment here differs from attention models? They surely resemble each-other though the alignment seems not that rich.


-  last but not least, I have found the language confusing. Some examples,
   -p2 bandwidth signal than the traditional label-only signal : I am very confused by how bandwidth comes to the picture and how this can be measured/justified

  - fig.1, what are \phi and \psi. paper never discussed these.

  - the authors say M is a tensor with 3dimensions. Then the marginalization before eq.1 uses M_{i,\cdot,\cdot} = 1 . What does this really mean?

---

> ### Public Comment · (anonymous) · 2018-11-14
> **Author Response to Reviewer 2**
>
> Thank you for the detailed comments.
>
> 1) C is not a binary map, but a continuous cost matrix where C_{i,j,k} tells us how well pixel i of the target image aligns to pixel j of reference image k. If you are viewing this as a binary matching problem, then M is the binary matching matrix and C (when normalized row-wise) is the continuous relaxation of M.
>
> To construct C_{i,j,k}, we take the hyper-column descriptor of pixel i in the target image \phi_i(I) and the hyper-column descriptor of pixel j in reference image k \phi_j(S_k) and compute the similarity between them using cosine similarity as the normalized dot product (below Equation 2). P(M_{i,j,k} | I_t, S) is inverse-exponentially related to C_{i,j,k} (Equation 2)
>
> 2) The contextual feature encoding of a point is the hyper-column representation of the point and not just the pixel colors. Hence, the system will not align two red points unless the contextual information is also in agreement.
>
> 3) Our approach is motivated by finding correspondences. We express the solution as an alignment between sampled points in the two images which we force the system to perform, aided by self-regularization. The label of the target image is the reference image that best aligns with the various parts of the target image.
>
> 4) [same response as to AnonReviewer3] Our method computes an approximate, sampled matching between two images, not an exact matching as computing an exact matching while enforcing locality constraints is computationally difficult. We hence make the independent point matching assumption. Given the full image and a point, let us assume there is an oracle that performs the matching. In this case, the independent point matching assumption is reasonable as the oracle will assign P_{i,j,k}=1 when the correct matching is done for a given pixel i and 0 otherwise. In our method, points are given increasing context as we use the features of higher layers, so given enough filters at each layer, an oracle would essentially have access to the full image using the hyper-column description of a particular point. Self-regularization trains the system to learn to perform the matching task of the oracle. Hence, we expect the approximate solution to be good enough to be able to perform the original task - one-shot learning, and still learning a reasonable matching.
>
> 5) Visual attention corresponds to learning a mask over the input where the model should pay attention and then expressing the feature descriptor of the image as a combination of the feature descriptors across the image according to the attention weight. If one were to directly augment Matching Networks [Vinyals et al., 2016] with attention, a comparison between two images would be performed in the feature embedding space corresponding to a single similarity score between the resulting image feature vectors. On the other hand, alignment corresponds to the matching of individual points in a manner that incorporates the context around the points. Thus the comparison between two images is computed as the sum of similarity scores of individual pixels (hyper-column representations). Feature vectors are not compressed to a single feature descriptor.
>
> 6) a) By bandwidth, we mean that only a single label is provided per target image for training in the traditional one-shot learning case (low bandwidth). With alignment, self-regularization provides a label per point corresponding to the index of each matched pixel, thus leading to rich set of labels (feedback) in addition to the original one-shot label (hence high-bandwidth). We will revise the terminology to make it more understandable.
>
> b)  \phi and \psi are discussed below equation 2 in Section 3.1. \psi is the hyper-column descriptor of the pixel using the encoding network while \phi is an additional embedding of \psi. When we are looking at symmetric encoding of the target and reference images, then \phi = \psi. However, we note that the reference and target images need not be the same (for example, we could try to align a 1-channel gray-scale image to a 3-channel color image). In this case, we need to project the two hyper-column pixel descriptors into the same space, which corresponds to \phi being an embedding of \psi.
>
> c) We meant to say M_{i,correspondingMatch(i)} = 1. M_{i, \cdot, \cdot} was a typo.

---

### Official Review · AnonReviewer1 · 2018-11-05
**A new way of learning key point correspondence which can reflect visual concept**

**Rating:** 7
**Confidence:** 4

**Review:**

This paper proposed a new way of learning point-wise alignment of two images, and based on this idea, one-shot classification and open-set recognition can be further improved. The idea is interesting. As a human, when we say two images are similar, we may compare them locally and globally in our mind. However, traditional CNN models do not make direct comparisons. And this work give a good direction to further improve this motivation.

The paper is well written and easy to understand.

For the experiments, MNIST, Omniglot and MiniImageNet are used to demonstrate the effectiveness of the proposed method. From Figure 2. we can see many interesting correspondences.

---

> ### Public Comment · (anonymous) · 2018-11-14
> **Author Response to Reviewer 1**
>
> Thank you for the review. Your summary captures the essence of our paper.

---

### Official Review · AnonReviewer3 · 2018-11-05
**Sound empirical study**

**Rating:** 6
**Confidence:** 2

**Review:**

The authors propose a deep learning method based on image alignment to perform one-shot classification and open-set recognition. The proposed model is an extension of Matching Networks [Vinyals et al., 2016] where a different image embedding is adopted and a pixel-wise alignment step between test and reference image is added to the architecture.

The work relies on two strong assumptions: (i) to consider each point mapping as independent, and (ii) to consider the correct alignment much more likely than the incorrect ones. The manuscript doesn’t report arguments in favour of these assumptions. The motivation is partially covered by your statement “marginalizing over all possible matching is intractable”, nevertheless an explanation of why it is reasonable to introduce these assumptions is not clearly stated.

The self-regularization allows the model to have a performance improvement, and it is considered one of the contribution of this work. Nevertheless the manuscript doesn’t provide a detailed explanation on how the self regularization is designed. For example it is not clear whether the 10% and 20% pixel sampling is applied also during self regularization.

The model is computationally very expensive and force the use of only 10% of the target image pixels and 20% of the reference images’ pixels. The complexity is intrinsic of the pixel-wise alignment formulation, but in any case this approximation is a relevant approximation that is never justified. The use of hyper column descriptors is an effective workaround to achieve good performance even though this approximation. The discussion is neglecting to argue this aspect.

One motivation for proposing an alignment-based matching is a better explanation of results. The tacit assumption of the authors is that a classifier driven by a point-wise alignment may improve the interpretation. The random uniformly distributed subsampling of pixels makes the model less interpretable.It may occur for example as shown in figure 3 where the model finds some points that for human interpretation are not relevant and at the same time these points are matched with points that have some semantic meaning.

---

> ### Public Comment · (anonymous) · 2018-11-14
> **Author Response to Reviewer 3**
>
> Thank you for the comments.
>
> 1) (i) Our method computes an approximate, sampled matching between two images, not an exact matching as computing an exact matching while enforcing locality constraints is computationally difficult. We hence make the independent point matching assumption. Given the full image and a point, let us assume there is an oracle that performs the matching. In this case, the independent point matching assumption is reasonable as the oracle will assign P_{i,j,k}=1 when the correct matching is done for a given pixel i and 0 otherwise. In our method, points are given increasing context as we use the features of higher layers, so given enough filters at each layer, an oracle would essentially have access to the full image using the hyper-column description of a particular point. Self-regularization trains the system to learn to perform the matching task of the oracle. Hence, we expect the approximate solution to be good enough to be able to perform the original task - one-shot learning, and still learning a reasonable matching.
>
>
> (ii) If we assume that (i) gets us close to oracle level performance, then log P_{i,j,k} -> 0 when the correct matching is found for pixel i and log P_{i,j,k} << 0 otherwise. The similarity between the hyper-column feature vectors is expected to be high (and is trained to be high with self-regularization). With the independence assumption, since the image alignment score = \sum_i max_{j,k} log P_{i,j,k}, we expect that score -> 0 when the matching is correct and score << 0 already when there are a few misalignments. We expect most of the probability mass to be concentrated around the correct alignment due to the hyper-column representation of points.
>
> 2) When it comes to self-regularization, it does not matter as much whether 10-20 sampling is used or if the full alignment is computed, as long as the target pixels are also selected in the reference set (for convenience of determining the self-alignment matching matrix). As the regularization is selected as the categorical loss, we have for an image I, S = {I}
>
> self-regularization loss = CrossEntropyLoss(P(M | I) , Identity) = \sum_i CrossEntropyLoss(P(M_{i,*,1} |I), 1_i)  where 1_i is a vector with 1 at position i and 0 everywhere else. In other words, we want P(M_{i,j,1}|I) = 1 if i=j and 0 otherwise. Here, P(M_{i,j,1} | I, S) is proportional to C_{i,j}(I,I) as in Equation (2).
>
> In our experiments, we used the 10% sampling during self-regularization as well. We could have sampled an additional 10% for the "reference" portion of the self-alignment, but this has marginal benefit as we are already contrasting the possible locations for a pixel across 10% of the image.
>
> 3) The justification for the sampling is that as we are performing independent alignment (ie the net alignment cost is the sum of the alignment costs of the individual pixels), a reasonable pixel sampling will well approximate the alignment score of the full image. If the images are misaligned (or can't be aligned), we expect a reasonable fraction of the pixels to have poor alignment scores.

---

### Official Review · AnonReviewer4 · 2018-12-12
**point wise embedding (+) with greedy matching (-)**

**Rating:** 7
**Confidence:** 3

**Review:**

Authors argue that using average (independent) greedy matching of pixel embedding (based on 4-6 layer cnn hypercolumns) is a better metric for one-shot learning than just using final layer embedding of a 4-6 layer cnn for the whole image.  Their argument is backed by outperforming their baseline and getting competitive results on few shot learning tasks. Their method is much more computationally heavy than the baseline matching networks. In order to make training feasible, in practice they train with 90% dropout of test pixels embedding & 80% dropout of reference pixels embedding.


The caveats:
-> Using hyper-columns is related to adding residual connections. The question remains how much performance can be gained by just adding residual connections (with dropout) to the matching networks and letting the network automatically (or with a probability) choose to embed higher layers or lower ones. Adding the residual connection and just comparing the final layer embeddings is a cleaner method than ABM which  provides a richer embedding than baseline and could potentially close the performance gap between ABM and final layer matching.

->It is strictly designed for one-shot learning. It does not benefit from few shots (extra shots) and the fact that these different shots are getting classified as the same label. Vinyal et al mitigates this shortcoming by adding the FCE. However FCE is not directly applicable anymore. Author’s don’t suggest any alternatives either. Their smaller gains (or even worse than baseline without self-regularization) in the 5-shot cases is an evidence of this shortcoming.

The fact that SNAIL (TCML Mishra et al. (2017)) consistently outperforms this method puts a question mark on the significance of this work. If it was computationally feasible, authors could have used SNAIL and replaced the 64 dimensional embedding of each picture with the 10-20% hypercolumns. Essentially due to computational costs authors are sacrificing a more thorough matching system (non-greedy) for a richer embedding and they don’t get better results.


On the other hand, authors may argue that the hyper-column matching is not just about performance, whereas it also adds interpretability to why two images are categorized the same. Illustrations like fig. 3 for example shows that the model is not matching semantically similar points and can be used to debug & improve the model. While understanding why a blackbox matching network is making a mistake and improving, is  harder.
It would have been nice if authors used this added interpretability in some manner. Such as getting an idea about a regularizer, a prior, a mask, etc. and improved the performance.

I would argue for accepting this paper for two reasons.
-> Given that they beat their baseline and  they get comparable performance to sota even with a greedy matching (min-pooling followed by average pooling), is impressive. Furthermore, it is orthogonal to methods like SNAIL if the computational cost could be resolved.

-> They not only provide which image is a match but how they are matched, which could be interesting for one-shot detection as well as classification.


Question: At test/validation: do you still only categorize with 10,20% samples or do you average the full attention map for all test pixels?


Nit: The manuscript needs several passes of proofreading, spell & grammar checking. A few examples in the first couple of pages:
-> The citing format needs to be fixed (like: LSTMsRavi, there should be () around citations).
-> are not incompatible: are compatible
->incomprehensible sentence with two whiles: ABM networks outperforms these other state-of-the-art models on the open-set recognition task while in the one-shot setting by achieving a high accuracy of matching the non-open-set classes while maintaining a high F1 score for identifying samples in the open-set.
-> add dots to the end of contribution list items.
-> we first look pairwise matchings: we first look at the pairwise matchings

---

### Meta-Review · Area_Chair1 · 2018-12-13
**Potentially interpretable few-shot learning algorithm.**

**Confidence:** 3
**Recommendation:** Reject

**Metareview:**

The reviewers are polarized on this paper and the overall feeling is that it is not quite ready for publication. There is also an interesting interpretability aspect that, while given as a motivation for the approach, is never really explored beyond showing some figures of alignments. One of the main concerns of the method’s effectiveness in practice is the computational cost. There is also concern from one of the reviewers that the formulation could result in creating sparse matching maps where only a few pixels get matched. The authors provide some justification for why this wouldn’t happen, and this should be put in a future draft. Even better would be to show statistics to demonstrate empirically that this doesn’t happen.

There were a number of clarifications that were brought up during the discussion, and the authors should go over this carefully and update the draft to resolve these issues. There is also a typo in the title that should be fixed.